# Pre-Interventional 3D-Printing-Assisted Planning of Flow Disrupter Implantation for the Treatment of an Intracranial Aneurysm

**DOI:** 10.3390/jcm11112950

**Published:** 2022-05-24

**Authors:** Guillaume Charbonnier, Panagiotis Primikiris, Benjamin Billottet, Aurélien Louvrier, Sergio Vancheri, Serine Ferhat, Alessandra Biondi

**Affiliations:** 1Interventional Neuroradiology Department, Besançon University Hospital, 25000 Besançon, France; pprimikiris@chu-besancon.fr (P.P.); sergiovancheri@gmail.com (S.V.); serine-ferhat@outlook.com (S.F.); biondi.alessandra@gmail.com (A.B.); 2Neurology Department, Besançon University Hospital, 25000 Besançon, France; 3Laboratoire de Recherches Intégratives en Neurosciences et Psychologie Cognitive, University of Bourgogne-Franche-Comté, 25000 Besançon, France; 43D Medical Printing Department, Besançon University Hospital, 25000 Besançon, France; bbillottet@chu-besancon.fr (B.B.); alouvrier@chu-besancon.fr (A.L.); 5Chirurgie Maxillo-Faciale, Stomatologie et Odontologie Hospitalière, CHU Besançon, 25000 Besançon, France

**Keywords:** aneurysm, 3D printing, interventional neuroradiology, stroke, web, contour

## Abstract

Intrasaccular flow disrupter devices (ISFD) have opened up new ways to treat intracranial aneurysms but choosing the correct size of ISFD can be challenging. We describe the first use of 3D printing to assist in the choice of ISFD, and we report an illustrative case. We developed a technique that uses preoperative angiography to make a plastic model of the aneurysm. We tested the deployment of different sizes of intrasaccular flow disruptor on the 3D model under fluoroscopy. The best devices were then used as the first-line strategy to treat the patient. The preoperative 3D printing helped in the successful selection of a first-line ISFD, which was not the one recommended by the manufacturer. Three-dimensional printing can provide interesting information regarding the treatment of intracranial aneurysms using ISFD. Further studies are needed to fully assess its benefits.

## 1. Introduction

In recent years, the introduction of the first intrasaccular flow disrupter devices (ISFD), the LUNA devices [1], has opened up a new way of treating intracranial aneurysms (IA), including those with a wide neck. Subsequently, Woven Endobridge (WEB, Terumo, Tokyo, Japan) [2] and CONTOUR (Cerus Endovascular, Fremont, CA, USA) [3] ISFDs have led to further significant progress in the field. These latter two devices are particularly useful in the treatment of challenging aneurysms. However, the choice of the appropriate ISFD type and size is crucial but is made difficult by the fact that irregularly shaped aneurysms may not always adapt as predicted by the manufacturer [4]. In addition, the intraprocedural testing of multiple devices increases the cost and duration of the procedure as well as the patient’s radiation exposure. Changing the device during the procedure also increases the number of manipulations, which could increase the risk of complications. The development of techniques to optimise and improve the procedure is therefore essential. In recent years, 3D printing has been developed within the medical field and has proved to be particularly useful in neurosurgical procedure simulation for intracranial aneurysms [5,6]. Although an increasing number of centres succeed in modelling 3D aneurysms, only one study has reported the use of the model to choose the size of an endovascular device, in this case, a flow diverter [7,8]. However, no method for using a 3D-printed model to assist in the choice of an intrasaccular flow disrupter has yet been described. In the present article, we present a new planning technique for IA endovascular treatment and report an illustrative case.

## 2. Materials and Methods

Our centre began studying 3D printing in the preprocedural planning of IA endovascular treatment in December 2020. Before presenting this case report we had a preliminary experience of 50 cases in order to finalize a standard procedure and master the technique before attempting to apply it to clinical practice. In this preliminary experience various problems emerged and were resolved, such as an incorrect choice of resin and the unavailability of flow disruptors of appropriate size. We present the different steps of the technique and describe an illustrative case in which the 3D printing allowed us to select the appropriate intrasaccular device, which was not the device recommended by the manufacturer.

### 2.1. Segmentation of the 3D Model

The first step of the preoperative preparation was the 3D segmentation of the aneurysm. We acquired images from the diagnostic 3D rotational angiography before the procedure, which used an Artis Q biplane angiography system (Siemens Healthineers, Erlangen, Germany). The 3D images were imported into the Mimics software (Materialise, Leuven, Belgium), version 23. To isolate the injected artery lumen volumes, we used thresholding, which allowed us to delete any volume below a certain density. The very high density obtained by intra-arterial injection of contrast medium during cerebral angiography made it easy to isolate the injected vessel from the bone and brain parenchyma. After applying the threshold, the artery lumen was isolated and we removed any remaining volume not in contact with the lumen. We then manually isolated the aneurysm. At this stage, the primary objective was to replicate the anatomic characteristics of the aneurysm as accurately as possible, including its implantation and the adjacent branches. The secondary objective was to create an artificial vessel system with entry and exit points to allow the flow circulation for the simulation in the angio suite. Based on our experience, the injection of the model was more efficient when the entry and exit points of the upstream and downstream vessels were parallel. This was for two main reasons: first, the parallelism allowed a better injection of contrast media, and second, it was easier to orient the connected tubes to the same container. For practical reasons, in order to create efficient watertight connections, we found it useful to extend the 3D model by adding 1–2 cm of a longitudinal tube at each entry and exit vessel point (Figure 1). During the above procedure, a senior neurointerventionist (GC) compared the 3D model to the 3D reconstructions obtained during the angiography. A layer around the artery volume was added in order to mimic the arterial wall. We chose a wall thickness of 1 mm to eliminate the internal structural support that is usually needed for plastic printing.

### 2.2. Production of the 3D Model

The models were then produced using a Formlabs FORM 2 3D printer and a photopolymerisation-based stereolithography process. The material used was a standard Formlabs Clear Resin (V4). We preferred not to use flexible resin for the model since during the initial attempts the flexible resin models were fragile and prone to liquid leaks. We printed stoppers to ensure that the entry and exit points were watertight. These were tailored to the catheter sizes we used. The average time for production was 90 min. Figure 2 shows the result of the 3D model as simulated by the computer (2A) and the 3D print (2B). The cost of the materials was very low (the resin cost EUR 0.40) but the 3D printer costs around EUR 3500, and one also has to consider the fact that a trained engineer spent approximately 2–3 h on each model.

### 2.3. Testing the 3D Model

The 3D model was tested in the same angio suite as used for the diagnostic angiography and endovascular treatment. We inserted approximately 1 cm of the Neuron MAX delivery catheter (Penumbra Inc., Alameda, CA, USA) through the entry point. The exit point was extended using plastic tubing, the distal end of which was attached to a plastic container that collected the water and contrast agent. The container was placed outside the range of the flat panel detectors in order to optimise the image quality, with or without digital subtraction. For the same reason, we used as little radiopaque material as possible and attached the model to the angio table using adhesive tape.

We then manually injected the contrast agent (IOMERON 300, Bracco Imaging, Milano, Italy) through the Neuron MAX into the model. Three-dimensional images were obtained using cone beam acquisitions. The absence of flow circulation in the model allowed a 3D acquisition with contrast medium staining without continual injection during the acquisition. Then, treatment positions selected from the 3D images were reproduced on the flat panels. Subsequently, the model was flushed with saline solution and the aneurysm was catheterized with the appropriate microcatheter and microwire. Once the different devices had been fully deployed, angiography runs were performed with and without digital subtraction.

The devices used during the 3D model test came from two sources. The manufacturer provided some unsterilized devices as models for demonstrations. Others were collected during patient procedures, from which devices were removed because they were considered unsuitable for the aneurysm. These devices were kept for 3D model tests after a visual inspection to exclude potential damage.

## 3. Results

In this case report we present the successful clinical application of the 3D printing technique described above in an illustrative case. A patient in their 50s with an unruptured left posterior inferior cerebellar artery (PICA) aneurysm was scheduled for an endovascular treatment with an intra-aneurysmal flow disrupter. We undertook pre-interventional planning using a 3D-printed plastic model. The diagnostic cerebral angiography of our patient revealed an irregular aneurysm at the origin of the left PICA (width 7.6 mm, height 3.6 mm, neck 5.8 mm) (Figure 3). The consensus of the multidisciplinary medical team meeting was to proceed with an endovascular treatment using a WEB or CONTOUR intrasaccular flow disrupter.

For this specific aneurysm with a smallest height of 5 mm and an average width of 5.5 mm (excluding the anterior bleb), the manufacturer’s recommendation was a WEB SL 6-4. The aneurysm neck of 3.8 mm and the mean equatorial plane diameter of 6.6 mm corresponded to a CONTOUR 7 or 9.

The diagnostic angiography was performed 3 months before the intervention. A 3D-printed replica of the aneurysm was constructed based on the diagnostic angiography, using the technique described above. One month before the intervention, we used the 3D model to test the four devices that seemed the most suitable for the treatment: WEB SL 6-3, WEB SL 6-4, CONTOUR 7 and CONTOUR 9. The resulting imaging of the different devices is presented in Figure 4.

The WEB SL 6-3 was considered the most suitable in terms of the exclusion of the aneurysm neck and preservation of the PICA’s origin (Figure 4A,B). To obtain an optimal result, we needed to adjust the deployment of the device. Instead of deploying the WEB in the middle of the aneurysm, it needed to be deployed slightly more proximally, towards the aneurysm neck, to create a proximal “plug”. It was not possible to deploy the WEB SL 6-4 with a satisfactory result. It either settled too deep (and therefore not blocking the neck on the PICA side, Figure 4C), or too low (therefore obstructing the vertebral artery flow (Figure 4D). The CONTOUR 7 remained too proximal, possibly compromising the flow in the PICA (Figure 4E). Finally, it was possible to deploy the CONTOUR 9 in a better position in terms of the exclusion of the aneurysm neck, but it remained too distal to the PICA and left a dog-ear neck remnant (Figure 4F). We therefore selected the WEB SL 6-3 as the first-line treatment device.

The intervention was performed one month after the simulation on the 3D-printed model. After the initial deployment, we manoeuvred the device into the desired position, which took a total of 11 min. The entire procedure lasted a total of 54 min from the acquisition of the 3D images to the detachment of the device. The total radiation dose was 9852 µGy·m². The patient did not experience any complications related to the procedure. The immediate postoperative angiography images are presented in Figure 5. An almost complete aneurysm occlusion was observed immediately after the procedure, with some contrast medium through the device and contrast stagnation in the dome.

## 4. Discussion

To our knowledge, this is the first study to describe the successful use of 3D printing for the preoperative planning of an intra-aneurysmal flow disrupter implantation procedure. Although the WEB-IT protocol included a new training program assisted by 3D modelling, which was sometimes based on preoperative imagery, no previous publications have shown the direct impact in device selection regarding size and type of flow disrupters [9]. The role of 3D preoperative planning has already been reported and partially explored in multiple fields of neurosurgery such as tumour resections, ventriculostomy and intracranial aneurysms [10]. Regarding neuroendovascular treatments, the 3D preoperative planning could help to reduce the number of devices tested, thereby reducing the operative manipulations, the procedure duration, the rate of complications, the radiation exposure, and the cost of the procedure. It could also be useful in endovascular training [6], as has already been done with virtual reality in open neurosurgery [11,12]. Furthermore, it has also been a useful tool for the patient’s education [13]. Finally, the development of our technique could be also adapted for intracranial stent implantation and other devices.

As the number of patients treated using a flow disrupter increases, it is important to better understand the sizing and develop methods to improve device selection. Three-dimensional software simulations have already been used in neurosurgery, associated with virtual and augmented reality [14,15]. A recent study demonstrated the utility of preoperative 3D software simulation for the selection of the WEB device [4]. It was a comparative study including 109 patients treated with WEB using computer assisted simulation in one centre and 77 patients in another centre, for whom computer assisted simulation was not used. Virtual simulation was associated with shorter intervention time, lower radiation dose, and lower number of WEBs deployed. Three-dimensional modelling in neuroendovascular treatments may also help to predict the deployment of the device in the aneurysm and the possibility of manipulating the device using the microcatheter to better orientate it so as to fit the geometry of the neck. Preoperative planning using a 3D model has significant advantages compared to a classic preparation procedure. First, there is a clearer visualisation of the device due to the absence of the skull bone, making it possible to work without subtracted angiography. Secondly, the model can be orientated to any angle for the best possible visualisation of the neck of the aneurysm. Additionally, the neurointerventional team is able to perform fluoroscopies and angiography series without exposing the patient to additional radiation. The use of 3D models in the angiography suite is also an easy and safe way to include junior team members for teaching and training purposes [6]. Finally, 3D-printed models can also be used to help explain the procedure to the patient at the preoperative consultation.

### Limitations

This study has certain limitations. Firstly, it is a case report that needs to be replicated in larger series. A randomised study would be appropriate to provide evidence on the utility of the technique. It could help to demonstrate that simulation has a direct impact on the procedural time, number of devices used, and the dose of radiation. The 3D models we print are not a perfect duplicate of reality. More specifically, we chose to print only the parent artery with the first curves in order to make this technique easier to apply and less time-consuming. A complete printing of the intracranial angiography, supra-aortic trunks, and aorta would permit a better anticipation of catheter navigation but does not seem necessary in order to predict the position of the flow disrupter in the aneurysm. Furthermore, the material used to produce the 3D model obviously behaves differently in terms of elasticity and friction compared to the human vessel wall. The improvement and further development of the materials in order to achieve more realistic 3D models should be advocated. Such models could use flexible resin with elastic properties calculated on the basis of real arteries. Finally, we did not detach the intrasaccular flow disrupter during the simulation as we tried to preserve it for reuse when it was not damaged. Detachment of the device could theoretically improve the fidelity of the simulation, but the flow disruptors are very expansive and thus cannot be purchased exclusively for presurgical planning. Three-dimensional printing devoted to medical practice is not currently easily accessible, even though an increasing number of centres are equipped with this relatively affordable technology. The main barrier is the time needed to master the required techniques even though it is not a steep learning curve.

## 5. Conclusions

Three-dimensional printing may be a useful tool for preoperative planning for some neurointerventional procedures in clinical practice. We reported here that it could be particularly useful for the selection of the flow disrupter size and type, which can often be challenging. As a result, 3D printing could have significant advantages in terms of reduction of procedure time, complication rates, radiation exposure, and costs, as well as in terms of patient education and the training of junior operators.

## Figures and Tables

**Figure 1 jcm-11-02950-f001:**
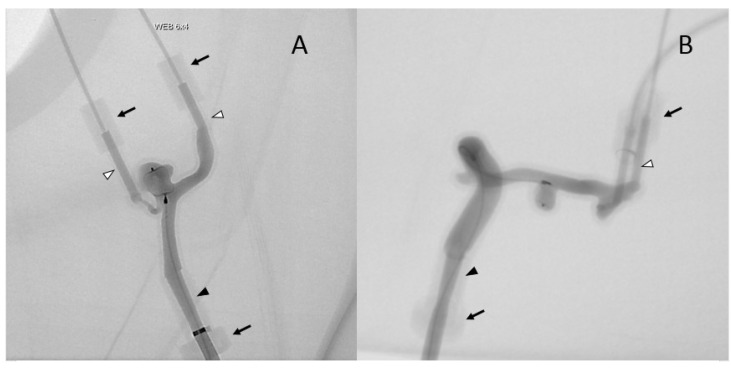
Injection of two 3D models (**A**,**B**) under fluoroscopy in the angio suite. Elongated point of entry of the vertebral artery (**A**) and internal carotid artery (**B**) (black arrowhead); elongated points of exit of the PICA and basilar trunk (**A**) middle cerebral artery (**B**) (white arrowheads) parallel to entry tube; soft waterproof plastic cylinders (black arrows).

**Figure 2 jcm-11-02950-f002:**
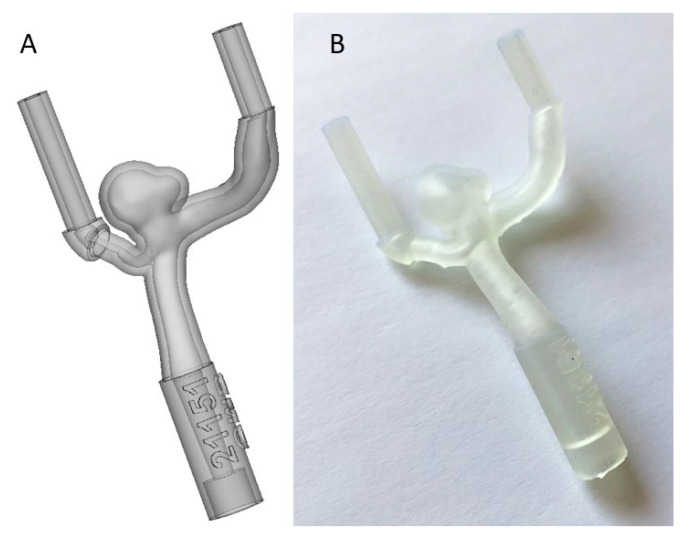
The result of the 3D segmentation (**A**) and the 3D final print (**B**).

**Figure 3 jcm-11-02950-f003:**
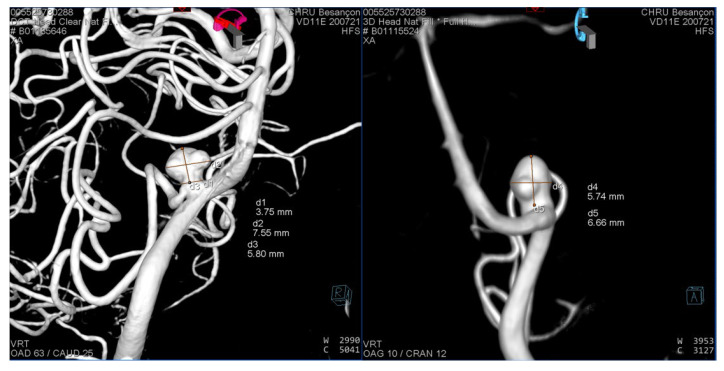
3D rotational angiography reconstruction of the left vertebral artery 3 months before the intervention.

**Figure 4 jcm-11-02950-f004:**
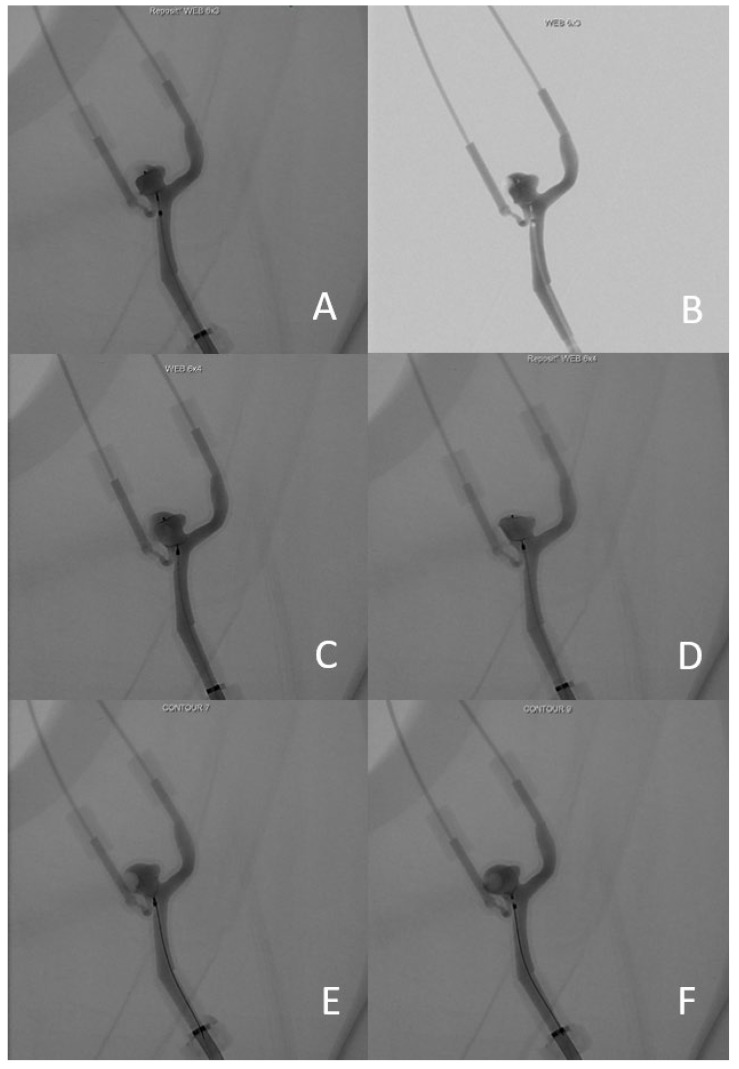
(**A**) angiography image of WEB 6-3 without subtraction; (**B**) angiography image of WEB 6-3 with subtraction; (**C**) WEB 6-4 position 1; (**D**) Web 6-4 position 2; (**E**) CONTOUR 7; (**F**) CONTOUR 9.

**Figure 5 jcm-11-02950-f005:**
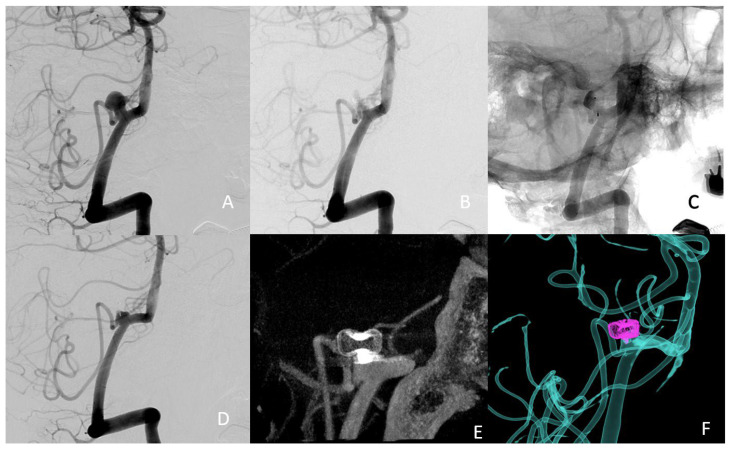
(**A**) preoperative treatment angle; (**B**) subtracted postoperative image; (**C**) nonsubtracted postoperative image, showing the stagnation of the contrast agent in the dome of the aneurysm, suggesting successful occlusion (**D**) 3-month follow-up treatment angle (**E**) 3-month follow-up 3D angio-CT (**F**) 3-month follow-up fusion imaging.

## Data Availability

All data from this study is available from the corresponding author upon reasonable request.

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
