# Peer review of "Pre-Interventional 3D-Printing-Assisted Planning of Flow Disrupter Implantation for the Treatment of an Intracranial Aneurysm"

_jcm, 2022, doi:10.3390/jcm11112950_

Round 1
Reviewer 1 Report
I would like to congratulate the authors. The paper is very interesting and innovative.
However, what I am not able to understand is whether it is a case report , so pre-procedural simulation of device implantation on one patient, or it is a description of a study on 50 patients.
If the latter then please provide the relevant results for this group.
Questions:
- why did you not print a wider segment of the arteries? To increase the realism of the catheter alignment
- don't you think that preparing the 3D model in an elastic form would increase the realism of the procedure? Formlabs has an elastic resin in their portfolio why did you not use it?
- what is the cost of 3D printing one model?
- what should be the future direction for preoperative simulation? E.g. adding a small pump to improve the realism of angiography? Do you think there is any preoperative way to confirm the exclusion of aneurysm flow? At this point, you are only relying on the morphological position of the device in the aneurysm
- Did you release the device in the 3d model or did you keep in on the delivery wire? Do you think it could impact the results?
Author Response
We would like to thank you for the kind remarks and your advice.
We would like to clarify that this a case report. Before presenting this case report we have previously worked on 50 patients in order to master the technique. The text has been modified accordingly in Material and Methods (line 60).
- The simulation would possibly be more accurate with the complete printing of the aorta and the supra-aortic trunks. When we did the first tests it appeared that the simulation was a lot more complicated and time consuming in terms of navigation without adding significant information in terms of the choice of the flow disrupter. Consequently, we simplified our technique so that it could be easily integrated in the normal activity of an Interventional Neuroradiology department and choose to print only the parent artery with the first curves as this seems sufficient in our experience in order to define the appropriate flow disruptor. This is a limitation that we now added to the manuscript (l.223)
- We indeed reported in the methods and limitation section that we used a non-flexible resin. We have tried multiple times to do the model with flexible resin but the results were worse as the model is more fragile and prone to liquid leaks. Therefore the injection under radioscopy was not satisfactory. We have added a specification based on your remark in the methods section (l.93).
- The cost is relatively low and has now been specified in Material and Methods, paragraph 2.2, line 99.
- From our point of view we have to develop high fidelity models. To achieve this objective we need not simply a flexible resin but a grade of flexibility similar to that of the artery a field which requires further research. This an essential point to achieve before seeking to simulate blood, pressure, systole-diastolic cycles and thrombosis. Regarding the use of intra-saccular flow disrupter the total exclusion of the aneurysm requires hours or days of stasis and this will remain a limitation of pre-operative planning and evaluation. The only objective of 3D printing for this study was to select the appropriate size of intrasaccular flow disrupter. This can already be very helpful as it can be difficult for irregular aneurysms. These points are now discussed l.229.
- The devices were not released as they are very expansive and we try to keep at least one from each size. The technique would indeed improve in fidelity if we detached the device. This limitation is now reported in l.234
Reviewer 2 Report
I read with great attention and interest the paper entitled "Pre-interventional 3D-printing-assisted planning of flow disrupter implantation for the treatment of an intracranial aneurysm"
I would congratulate the Authors for their excellent work and their idea about a new methods for preoperative planning in case of cerebral aneurysm
I think 3D printed model could represent an important field in modern medicine
I would only suggest the Authors to insert a proper reference for every single affirmation reported in Introduction and Discussion
Moreover, I would suggest to change the article reporting results of all treated patients: particularly about model accuracy
Author Response
We would like to thank you for you kind remarks.
- We added references as requested (please see added references 10-11-12-13-14-15)
- It would indeed be very interesting to publish data from a wider cohort.
We would like to clarify that this a case report. Before presenting this case report we have previously worked on 50 patients in order to master the technique. The text has been modified accordingly in Material and Methods further explaining this point (line 49).
Reviewer 3 Report
This is a very current topic, but some points need to be addressed:
- Authors discussed about the 3D-printing-assisted planning in their paper, but no figures of a 3D-printing object was present. Please add it to the manuscript.
- Lines 49-50: "Our preliminary experience including 50 cases allowed 49
us to finalize a standard procedure.." but after in the text, authors does not report any data about those 50 cases. So is this a case report ? - Lines 116-117: Similarly, results section starts with "The diagnostic cerebral angiography of our patient revealed an irregular aneurysm at the origin of the left PICA..." This is not well. Please revise the start of this paragraph.
- Lines 165-167: "use of 3D printing for the pre-operative planning" what are the main advantages of this procedure? Please discuss. Look at: doi: 10.3390/ijerph19031719 -- doi: 10.3171/2021.5.FOCUS2210 -- doi: 10.3390/ijerph19084
- Lines 175-177: "A recent study demonstrated the utility of preoperative 3D software simulation for the selection of the WEB device [9]". Improve this point
- Lines 175-176: "preoperative 3D software simulation..." These 3D software simulations are width use in neurosurgery in association with virtual/augmented/mixed reality. Discuss more. doi: 10.1016/j.xnsj.2021.100063 -- doi: 10.3390/ijerph18199955
- Lines 191-192: "Firstly, it is a pilot study that needs to be replicated in larger series" In the way the work is set up, it does not appear to be a pilot study. This sentence is not correct.
- Lines 192-193: "A randomised study would be appropriate to provide evidence about the utility of the technique" This sentence is not relevant. What did authors mean?
- In conclusion section, please report what this paper add new to the literature.
Author Response
We would like to thank you for your pertinent remarks.
- A figure of the 3D segmentation and a picture of the printed 3D model have been added to the manuscript. L.103
- It would indeed be very interesting to publish data from a wider cohort.
We would like to clarify that this a case report. Before presenting this case report we have previously worked on 50 patients in order to master the technique. The text has been modified accordingly in Material and Methods (line 49).
- The beginning of the paragraph has been modified in order to clarify that we present the results of an illustrative case. We have also added clinical information concerning the case reported (l.129).
- We detailed this section and added more references, including the first DOI provided by the reviewer. Unfortunately, the 2 other DOI didn’t seem to work. L.187-200
- We reported the main results of this specific study l.201.
- The text has been accordingly enriched in discussion l.187-200 with the 2 suggested citations.
- We have replaced « pilot study » with « case report » L.219
- We plan to start a randomized study with a control group who will not benefit from the 3D printing-assisted-simulation. The objective will be to demonstrate the advantages regarding the procedural metrics in the simulation group. We have detailed why we believe that a randomized controlled study could be useful L.220.
- In conclusions l.242 we highlight the possible advantages of this technique for the treatment with intrasaccular flow disrupters.
Round 2
Reviewer 1 Report
Congrats to the authors
very interesting article
Reviewer 2 Report
No further comments
Reviewer 3 Report
Authors solved all my criticisms.